# Use of the Cover-Lifting Technique in Mandibular Cemento-Ossifying Fibroma Excision to Preserve the Inferior Alveolar Nerve

**DOI:** 10.3390/medicina57121383

**Published:** 2021-12-19

**Authors:** Juan-You Qiu, Kuan-Min Huang, Nan-Chin Lin

**Affiliations:** 1Department of Oral and Maxillofacial Surgery, Show Chwan Memorial Hospital, Changhua 515, Taiwan; amykevin0726@gmail.com; 2School of Dentistry, China Medical University, Taichung 404, Taiwan

**Keywords:** cemento-ossifying fibromas, inferior alveolar nerve, 3D-printed cutting guide

## Abstract

Cemento-ossifying fibroma (also known as ossifying fibroma or cementifying fibroma) is a benign osteogenic neoplasm. Pain and paresthesia are rarely associated with cemento-ossifying fibroma; thus, nerves must be preserved during excision. With the advent of computer-aided techniques, the use of virtual surgical planning and a customized template can improve the precision of resection and reconstruction, reduce operating time, and improve postoperative outcomes. In this report, we describe a case of cemento-ossifying fibroma in a female patient who underwent segmental mandibulectomy and reconstruction with an iliac bone graft. Additionally, we describe a simple and effective way to preserve the inferior alveolar nerve.

## 1. Introduction

Cemento-ossifying fibromas are classified as fibro-osseous lesions, which also include fibrous dysplasia and various types of cemento-osseous dysplasia. Cemento-ossifying fibroma (also known as ossifying fibroma or cementifying fibroma) is a benign osteogenic neoplasm that is well demarcated, and composed of fibrous tissue that contains varying amounts of mineralized materials resembling bony trabeculae, cementum, or both [1]. It manifests as an asymptomatic slow-growing lesion, but it may cause an obvious facial deformity if left untreated [2]. It is more common in women than in men, and the molar–premolar region of the mandible is the most common location. Pain and paresthesia are rarely associated with cemento-ossifying fibroma [3].

This tumor can be treated with conservative enucleation and curettage. Nevertheless, for large tumors that have destroyed considerable bone, resection and reconstruction with a bone graft may be essential [4]. Traditional surgical resection with high-speed burs and reciprocating saws may damage the inferior alveolar neurovascular bundle, which results in the numbness of the lower lip and chin after the surgery [5]. Kuriakose et al. [6] described a technique for preserving the inferior alveolar nerve (IAN) during segmental mandibulectomy for nonmalignant lesions. They traced the path of the IAN on conventional panoramic images with little intraoperative guidance. However, the nerve canal was often displaced because of pressure exerted by the tumor; thus, accurate localization was limited [5]. Computer-aided techniques have enabled virtual surgical planning (VSP) and the customization of templates, which improve the precision of resection and reconstruction, reduce operating time, and improve postoperative outcomes [5].

We describe a case of cemento-ossifying fibroma in a female patient who underwent segmental mandibulectomy and reconstruction with an iliac bone graft, and we describe a simple and effective way to preserve the IAN.

## 2. Case Presentation

An asymptomatic bony lesion in the left lower jaw was found in a 44-year-old Taiwanese woman at a local dental clinic, when a pre-dental implant radiographic survey was performed. The doctor then referred her to the Department of Oral and Maxillofacial Surgery of Show Chwan Memorial Hospital, Changhua, Taiwan. She had no drug allergies or systemic disease, and denied any history of smoking, drinking alcohol, or chewing betel nuts. Extra-oral examination revealed a firm, non-removable mass in the left lower jaw. No facial numbness was present. Intra-oral examination revealed a firm, non-tender swelling in the gingiva of the left lower jaw. Panoramic radiographs and computed tomographic scans revealed a mixed radiolucent–radiopaque, expansile bony lesion (3 × 3 cm) extending from the mesial root of the lower left first molar to the left mandibular angle (Figure 1). The distance from the bottom of the lesion to the inferior border of the mandible was less than 1 cm. An incisional biopsy was performed via the transoral route, which indicated a benign tumor as per findings of a fibro-osseous lesion. Surgical intervention was indicated, and so the patient was admitted to our outpatient surgical ward for further management. Due to the short distance from the tumor to the inferior border of the mandible, segmental resection instead of marginal resection was recommended [7]. The treatment plan included wide tumor excision with segmental mandibulectomy, preservation of the IAN, reconstruction with an iliac crest bone graft, and internal fixation and intermaxillary fixation (IMF).

VSP and customized three-dimensional (3D) printing was used for the treatment planning of this patient. To facilitate the surgery, a mandibular cutting guide and an iliac cutting guide were fabricated for the patient (Figure 2A,B). In surgery, the intraoral and submandibular approaches were used to access the tumor. First, the lower left first molar was extracted, and the cutting guide was temporarily fixed to the mandible with mini-screws. On the cutting guide, the route of the IAN was marked in and out of the tumor. A piezoelectric saw was used to perform the osteotomy on the proximal and distal sides of the tumor, according to the cutting guide, and the osteotomy was also completed to enable access to the IAN in and out of the tumor, above which lay the actual path of the IAN. Next, using the cover-lifting technique (Figure 2C), the nerve bundle was preserved and finally, the tumor was resected after the osteotomy was completed. 

Simultaneously, an incision was made in the left iliac region. After dissecting layer by layer, the anterior iliac crest was identified. Using the iliac cutting guide, piezoelectric surgery was performed similarly to the technique described previously. A saw and a chisel were used to section the bone block (3 × 5 × 2.5 cm) and then the iliac bone graft was inserted into the mandibular bony defect, and fixed with mini-plates and mini-screws. The shape of the iliac bone graft coincided with that in the pre-operative plan. IMF was performed for the patient to stabilize the maxillomandibular position (Figure 3).

After surgery, the wound showed no oozing and no sign of infection, and the patient was discharged 1 week later. A histopathological examination of the specimen sections revealed dense lamellar bone containing an unencapsulated tumor, composed of variable amounts of osteoid, woven bone, and basophilic cementum-like calcifications in a fibrous stroma (Figure 4). These findings confirmed the diagnosis of cemento-ossifying fibroma. At the 6-week follow-up appointment, the patient’s occlusion was stable, and the IMF was removed. The extent and degree of numbness in the lower lip and chin after the surgery subsided progressively, which indicated good healing.

## 3. Discussion

According to a meta-analysis conducted by Gou et al. [7], marginal mandibulectomy should be reserved for malignant tumors with superficial or no invasion of the cortical bone. For malignant or benign tumors that invade the medullary bone, or the lower border of the mandible by less than 1 cm, segmental mandibulectomy may be preferred. In the patient, the distance from the deepest part of the neoplasm to the inferior border of the mandible was less than 1 cm; thus, we performed segmental resection. According to the “6 cm rule” described by Marschall et al. [8], mandibular defects larger than 6 cm should be reconstructed with vascularized grafts, whereas nonvascular bone grafts should be used for smaller defects. The patient’s tumor was approximately 3 × 3 × 2.5 cm, and so, we chose a nonvascular iliac bone graft for reconstruction.

Cover-lifting used in this case—pre-operative VSP, the use of patient-specific 3D-printed guides and piezoelectric instruments in both the mandible and the iliac area for the osteotomy, the removal of lateral cortical bone with chisel and saw, separation of the IAN from the tumor, and segmental mandibulectomy—enabled us to preserve the IAN successfully.

Ricotta et al. [9] used a 3D-printed patient-specific surgical guide for ablation of a benign tumor to prevent IAN injury. A mark on the guide’s outer surface indicated the position of the IAN, and their template depicted the path of IAN from the mental foramen. Using the template and a piezoelectric saw, they performed osteoplasty of the outer mandibular cortex. Then, they removed the cortical bone outlined on the template according to the entire path of the IAN in the mandible. They gently removed the IAN from the mandibular canal, thereby preserving the nerve. By contrast, we removed the entire lateral cortical bone, extending beyond the IAN pathway to the lower border of the mandible. The cover-lifting technique could save time, and for some benign lesions, the removed cortical bone could be repositioned after the surgical ablation.

To access a benign mandibular tumor, Liu et al. [10] used sagittal split osteotomy and VSP, in which they identified the precise pathway of the entire IAN canal and the location of the tumor. After excising the benign tumor, they fixed the osteotomy segments with screws and a plate. Both the outer cortical bone and the IAN were preserved. In comparison, if used for cystic lesions, a cover-lifting technique could better preserve the continuity of the mandible. Moreover, IMF could be avoided.

Huang et al. [5] also designed a 3D digital template to preserve the IAN in reconstruction. The border of the mandibular benign lesion and the IAN canal were outlined on the template; during osteotomy, the IAN bundle was separated and protected accurately. Their digital template fitted well on the mandibular surface. In treating a case of ossifying fibroma, Chung et al. [11] fabricated a customized cutting guide to preserving the IAN, and they used a combination of VSP and piezoelectric surgery during the extirpation of the tumor.

## 4. Conclusions

To conclude, VSP with a 3D-printed cutting guide can improve the accuracy of resection and reconstruction and shorten the operating time. Furthermore, the cover-lifting technique can help preserve the IAN, thereby improving the patient’s quality of life postoperatively. However, we need more evidence to support the reliability of the cover-lifting technique. 

## Figures and Tables

**Figure 1 medicina-57-01383-f001:**
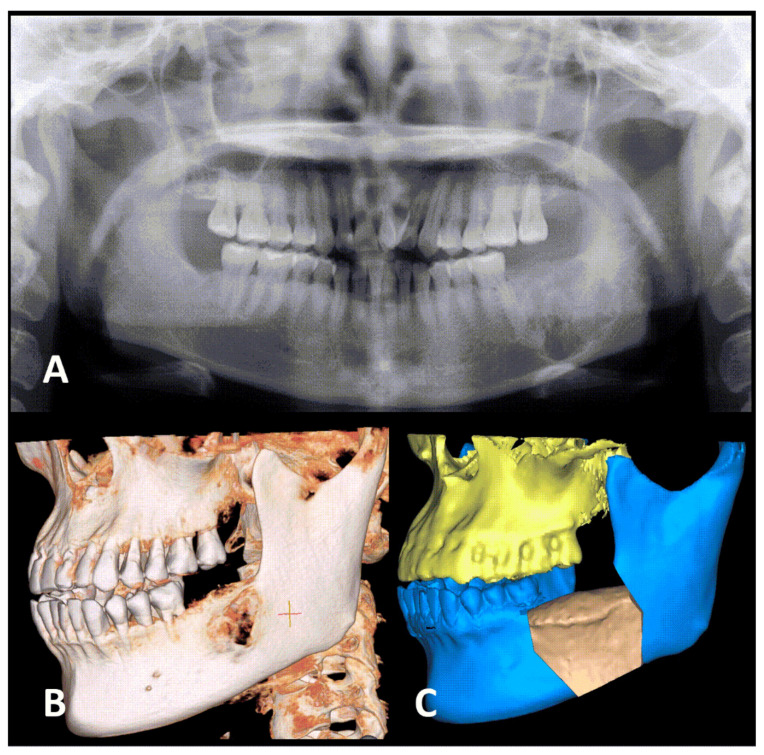
Pre-operative images of the jaw. (**A**) The panoramic radiograph, revealing a mixed (radiolucent–radiopaque) bony lesion (3 × 3 cm) in the posterior region of the left mandible region. The lesion was very close to the inferior border of the mandible. (**B**) Pre-operative three-dimensional image (left-sided view), showing the expansile tumor. (**C**) Pre-surgery plan to simulate the iliac bone graft reconstruction model.

**Figure 2 medicina-57-01383-f002:**
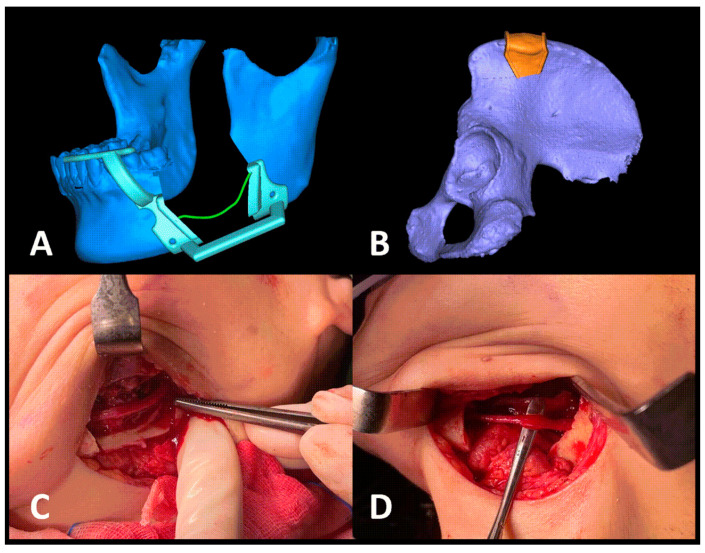
Three-dimensional virtual surgical plan for (**A**) a customized mandibular cutting guide and (**B**) an iliac cutting guide. (**C**) The appearance of the inferior alveolar nerve and tumor after mandibular the cortical bone block was removed. (**D**) Segmental mandibulectomy with preservation of the inferior alveolar nerve.

**Figure 3 medicina-57-01383-f003:**
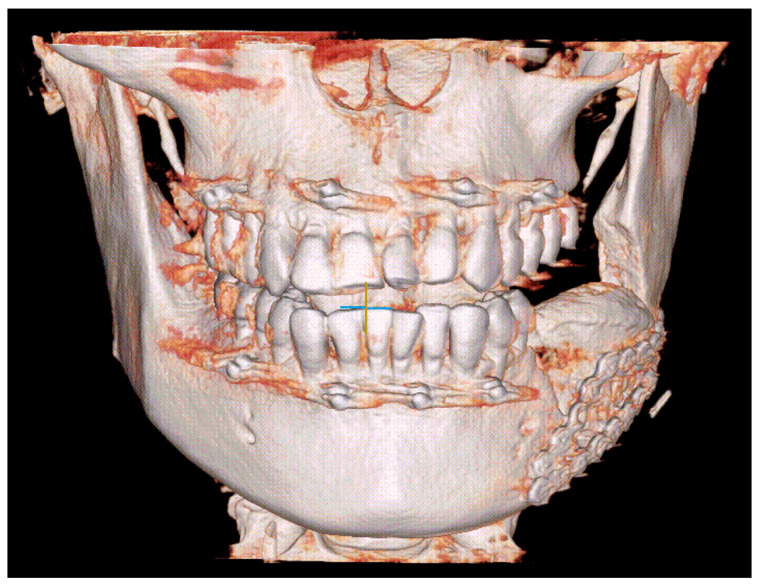
Post-operative computed tomographic reconstructed model, showing the iliac bone graft fixed with mini-plates and mini-screws according to the pre-operative plan, and intermaxillary fixation was performed.

**Figure 4 medicina-57-01383-f004:**
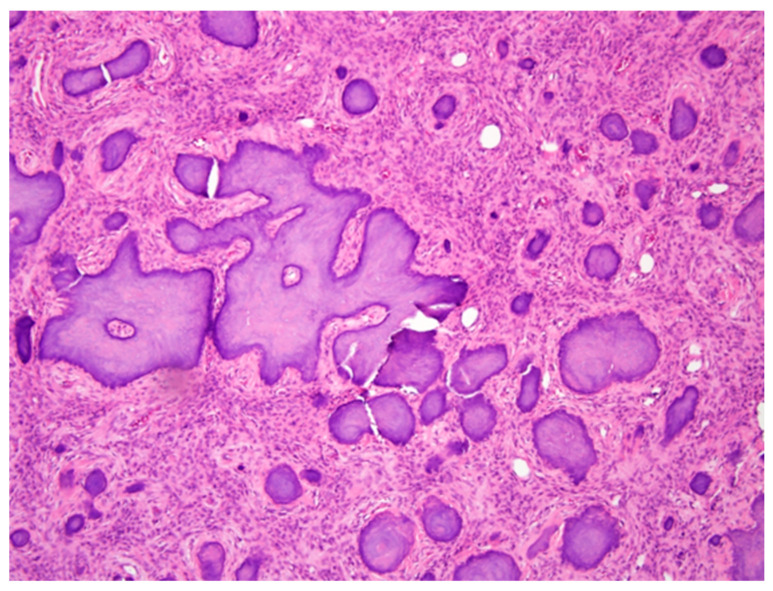
Histopathological micrograph, revealing curvilinear trabeculae formed by cementum-like calcifications in a fibrous stroma. The final diagnosis was cemento-ossifying fibroma.

## Data Availability

Not applicable.

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
