# Peer review of "Use of the Cover-Lifting Technique in Mandibular Cemento-Ossifying Fibroma Excision to Preserve the Inferior Alveolar Nerve"

_medicina, 2021, doi:10.3390/medicina57121383_

Round 1
Reviewer 1 Report
The study seems to be an interesting one with modified technique for removal of benign lower jaw lesions. In the discussion part, it has been mentioned that the technique is better from the one used by Liu et al as IMF could be avoided but in your case presentation, IMF was performed for 6 weeks to reinforce stabilization. The quality of life would be better if IMF would have been avoided or performed for a shorter period of time. This needs explanation.
Secondly with this combined intra-oral and extraoral approach, how did you manage any inadvertent injury to the marginal mandibular nerve as I suppose there is a lot of push and pull while retraction and putting cutting guide and fixing the screws. This needs some elaboration.
Still, few more rationale and scientific points need to be added to convince the readers that why this approach should be adopted from the rest mentioned in the discussion part.
Author Response
We sincerely thank the editor and all reviewers for their valuable feedback that we have used to improve the quality of our manuscript. The reviewer comments are laid out below in italicized font and specific concerns have been numbered. Our response is given in normal font and changes/additions to the manuscript are given in ”Track changes”.
Please see the attachment.

Reviewer 2 Report
Dear Authors,
The paper is novel & well written. The authors' have put good efforts in writing this paper.
All the best,

Author Response
We sincerely thank the editor and all reviewers for their valuable feedback that we have used to improve the quality of our manuscript. The reviewer comments are laid out below in italicized font and specific concerns have been numbered. Our response is given in normal font and changes/additions to the manuscript are given in ”Track changes”.
Thank you very much for your comments. I have revised the manuscript accordingly.
